# Evidence of Intragenic Recombination in African Horse Sickness Virus

**DOI:** 10.3390/v11070654

**Published:** 2019-07-18

**Authors:** Harry G. Ngoveni, Antoinette van Schalkwyk, J.J. Otto Koekemoer

**Affiliations:** 1Onderstepoort Veterinary Institute, 100 Old Soutpan road, Onderstepoort 0110, South Africa; 2Department of Veterinary and Tropical Diseases, Faculty of Veterinary Sciences, University of Pretoria, Onderstepoort 0110, South Africa

**Keywords:** African horse sickness virus, orbivirus genome, virus recombination, dsRNA virus

## Abstract

Intragenic recombination has been described in various RNA viruses as a mechanism to increase genetic diversity, resulting in increased virulence, expanded host range, or adaptability to a changing environment. Orbiviruses are no exception to this, with intragenic recombination previously detected in the type species, bluetongue virus (BTV). African horse sickness virus (AHSV) is a double-stranded RNA virus belonging to the *Oribivirus* genus in the family *Reoviridae*. Genetic recombination through reassortment has been described in AHSV, but not through homologous intragenic recombination. The influence of the latter on the evolution of AHSV was investigated by analyzing the complete genomes of more than 100 viruses to identify evidence of recombination. Segment-1, segment-6, segment-7, and segment-10 showed evidence of intragenic recombination, yet only one (Segment-10) of these events was manifested in subsequent lineages. The other three hybrid segments were as a result of recombination between field isolates and the vaccine derived live attenuated viruses (ALVs).

## 1. Introduction

African horse sickness virus (AHSV) causes a non-contagious, but highly infectious arthropod-borne disease of equids known as African horse sickness (AHS). In susceptible horses, the disease causes severe morbidity and mortality, while mules, donkeys, and zebra are less affected and considered to be vertebrate reservoirs [1,2]. The virus is arthropod-borne (arbovirus), transmitted from one vertebrate host to another through vector-competent midges feeding on host blood as a protein source [3]. Approximately 30 species of midges belonging to the genus *Culicoides* have been associated with AHSV transmission and replication [4]. The World Organization for Animal Health (OIE) has listed AHSV as a notifiable disease, due to its devastating economic effects and ability to rapidly spread from endemic to non-endemic regions [2]. In South Africa, annual outbreaks have a devastating economic and social impact due to direct losses as well as to the restrictions in movement of equids locally and internationally [5]. 

African horse sickness virus particles are non-enveloped, icosahedral, and consist of a double capsid layer containing the genome composed of ten linear, double-stranded RNA (dsRNA) segments [6,7]. The genome segments range in size from 3965 to 756 bp and are identified as segment-1 to segment-10 based on their decreasing molecular mass [7]. The segmented dsRNA genome codes for seven structural (VP1-7) and four non-structural proteins NS1, NS2, NS3/NS3A, and NS4 [8,9,10,11]. Nine antigenically distinct serotypes of AHSV (AHSV 1–9) have been identified, with the outer capsid proteins VP5 and, predominantly, VP2, contributing to this distinction [12,13,14,15].

Genetic recombination has no restrictions and can take place in segmented and non-segmented viruses, including the orbiviruses such as blue tongue virus (BTV) [16,17,18]. The frequency of recombination varies significantly among different virus classes, with the lowest rate observed in negative-sense RNA viruses [19,20]. Recombination generates new genetic combinations from the cross-over between different parental molecules [21]. The progeny virus could have increased virulence, an extended host range, the ability to evade the host immunity, or resistance to antivirals [18]. It is essential for two or more viruses to infect the same host cells, in order for their genetic material to be exchanged. The frequency of exchange between different viral genomes depends on the frequency at which co-infection of a host cell occurs [22]. The probability of co-infection is dependent on co-circulation of different strains at a specific time and geographical area, the prevalence of the population in the host as well as the co-infection rate [23].

A study by Woods [24] reported on the occurrence of recombination within genome segments of group A rotaviruses but the only evidence thus far for similar events in the orbiviruses has been reported in BTV [17]. Chimeric genes generated through intragenic recombination in BTV have been demonstrated, and recombination in these genes were shown to contribute to the genetic diversity of BTV. There are no reports concerning the effect of recombination, aside from reassortment, on AHSV genome diversity [25]. In order to determine the influence of homologous recombination on AHSV evolution, we analyzed each individual genome segment of more than 100 complete AHSV genomes.

## 2. Materials and Methods

### 2.1. Data Sets

Complete genome sequences of more than 100 AHSV isolates were used to analyze genomic recombination. The majority of the viruses were collected between 1933 and 2014 in Sub-Saharan Africa at the OIE World Reference Centre for AHSV and BTV at the Agricultural Research Council–Onderstepoort Veterinary Institute, South Africa. The complete genomes of 87 isolates were determined for this study (KP939368–KP940236 and KP009621–KP009790), while the sequences of an additional 14 were extracted from GenBank [26]. Sequences from the Onderstepoort Biological Products commercial African horse sickness attenuated live virus (ALV) vaccine were included (KT030330–KT030359 and KT715601–KT715640) [27,28]. 

### 2.2. Generating Complete Genome Sequences of African Horse Sickness Viruses

Monolayers of baby hamster kidney (BHK21) cells cultured in Dulbecco’s Modified Essential Medium (DMEM) (Gibco; Thermo Fisher Scientific, Waltham, USA) supplemented with 5% (*v*/*v*) foetal bovine serum (FBS), L-glutamine (200 mM) and 25 µg/mL Penicillin-Streptomycin-Amphotericin B (PSA) (Gibco) were infected with freeze-dried individual AHSV isolates (Appendix A). Total dsRNA was extracted and cDNA synthesized and individual genome segments amplified using the loop-mediated approach, previously described by Potgieter et al. [29]. Purified PCR products were sequenced using Nextera DNA Sample Preparation Kit (Illumina, Inc., San Diego, CA, USA) with an Illumina MiSeq instrument (Illumina, Inc.) at Inqaba Biotec (Pretoria, South Africa). Between 250 and 500 Mb of data was obtained for each isolate and this was imported into CLC Genomics Workbench v7.0 (CLC bio, Aarhus, Denmark). The reads were trimmed and low-quality score data was removed. Contigs were generated by either de novo assembly or mapping against known references. These contigs were used to assemble ten complete genome segments per sample. The final consensus sequence of each African horse sickness virus genome segment was submitted to GenBank and accession numbers assigned (KP939368–KP940236).

### 2.3. Phylogenetic and Recombination Analysis

An alignment of each of the ten genome segments were created using ClustalW as implemented in the MEGA v6 software [30]. The model of nucleotide substitution that best fits each data set was determined using the maximum likelihood model selection in MEGA v6 software [30]. Consequently, the data were analyzed under a General Time Reversible (GTR) model of nucleotide substitution with gamma distribution (T4). Maximum-likelihood phylogenetic trees were built by MEGA v6, using Bootstrapping of 1000 replicates to assess the robustness of a tree.

In order to detect evidence of intragenic recombination, the individual segment alignments were analysed using different methods described in the Recombination Detection Program (RDP) v4.39 package [31]. The RDP v4.39 was set to default, and a large window sizes of 200 nucleotides were used. The aligned sequences were analysed using a bootscan/rescan recombination test [32], MAXCHI [33], GENECONV [34], CHIMAERA [35], and the SISCAN method [36]. These different recombination tests were performed under the assumption that the data sequences could be used in a circular form and each method had a Bonferroni correction p-Value of 0.05. Potential recombinants were further described using SimPlot [37].

## 3. Results

Individual alignments of each of the ten virus genome segments were used to generate separate phylogenetic trees. Significant incongruences were detected between the tree topologies representing the ten genome segments, suggesting that recombination influences AHSV evolution. In order to detect evidence of intragenic recombination, the individual segment alignments were analyzed using different methods described in the RDP package. Potential recombination events were predicted in segment-1, segment-6, segment-7, and segment-10. These included both single and double cross-over events. Each putative recombination event was subsequently analyzed and will be discussed individually.

### 3.1. Segment-1 (VP1)

Phylogenetic analysis of the AHSV sequences divided segment-1 into two uneven major groups (Figure 1). The small group (B) constitute OIE-reference strains of serotype 1, 2, 3, 4, and 7 as described by Potgieter et al. [26], the respective ALVs of the same serotypes [27,28] as well as samples from serotype 1, 2, 4, 5, and 7 isolated prior to 1960 [26]. The remainder of segment-1 sequences all clustered into the larger group (A). A limited number of sequences belonging to the major group (A) are presented in Figure 1A. Results from the RDP package analysis predicted a double cross-over event (positions: 1524 and 1864) in Segment-1 of isolate AHSV-3_2_89. Phylogenetic trees constructed using Maximum-likelihood analysis inferred that virus KP93482_AHSV-3_2_89 grouped within the large group (A), while the minor parental sequences were from the small group (B) (Figure 1A). Statistical support of methods used in the RDP package, produced average *p*-values of RDP = 6.47 × 10^−20^, GENECONV = 3.37 × 10^−15^, BootScan = 2.903 × 10^−21^, MaxChi = 1.45 ×10^−8^, SiScan = 1.83 × 10^−8^, and 3Seq = 5.64 × 10^−11^ (Figure 1B). A similarity plot was constructed by using KP939933_AHSV-7_67_99 as the major parent and KP939479_AHSV-3_13_63 as the minor parent. Since KP939479_AHSV-3_13_63 was originally used in the generation of the live attenuated vaccine of serotype 3, the sequence of the corresponding ALV (KT030340_AHSV-3_OBP-116_1998, [27]) was included in the similarity plot. KP93482_AHSV-3_2_89 shared higher sequence identity to wild-type virus sequence KP939933_AHSV-7_67_99 before (98.9%) and after (99.8%) the two predicted breakpoints, than within (89.5%). In contrast, KP939479_AHSV-3_13_63 had a higher percentage sequence identity between the breakpoints (97%), than before (89.6%) and after (89.2%) to KP93482_AHSV-3_2_98 (Figure 1C). Maximum likelihood phylogenetic analysis of positions 1850–1510 and 1510–1850 produced incongruent phylogenetic trees. Analysis of the concatenated regions before and after the breakpoints (1850–1510), indicated that KP93482_AHSV-3_2_89 will cluster in the same large group as the major parental sequence (Figure 1D). When the region between the breakpoints was analyzed, KP93482_AHSV-3_2_89 clustered with the minor parental sequences in the small group (Figure 1E). Recombination within region 1524 to 1864 of KP93482_AHSV-3_2_89 with KP939479_AHSV-3_13_63 resulted in 34 synonymous substitutions compared to KP939933_AHSV-7_67_99 (data not shown).

### 3.2. Segment-6 (VP5)

Phylogenetic analysis of all the segment-6 AHSV sequences revealed that sequences of serotypes 1, 2, 4, 7, and 8 clustered together within individual serotypes, while serotypes 3, 5, 6, and 9 are combined within a single cluster (Figure 2A). A possible double-crossover event was identified relative to sequence KP939529_AHSV-3_DG25324_14 at positions 496 and 962 in comparison to vaccine ALVs KT030344_AHSV-3_OBP-116_1998 and KT15635_AHSV-8_OBP-252_1998. Significant statistical support for the events between 496 and 962bp had a *p*-value using Maxchi of *p* = 3.85 × 10^−5^, Chimaera (*p* = 5.49 × 10^−5^) and SiSscan predicted *p* = 2.30 × 10^−9^ (Figure 2B). AHSV-3_DG25324_14 was isolated in the Gauteng province of South Africa in 2014 and had a segment-2 identical to the ALV vaccine of serotype 3 (data not shown). Maximum-likelihood analysis of selected AHSV segment-6 sequences, indicated that KP939529_AHSV-3_DG25324_14 clustered between the major and minor parental sub-groups (Figure 2A). The major parental sub-group composed of sequences from serotype 3, 6, and 9, while the minor parental sub-group contained sequences from serotype 5 and two reassortments from serotype 8, including the ALV vaccine (KT15635_AHSV-8_OBP-252_1998). Recombination predictions in RDP suggested that any of AHSV-5 serotype sequences were the parent between base pairs 496 to 962, with reference sequence AHSV-5_30_62 sharing 92.2% sequence identity within this region comparing to 87.9% and 86.1% for the flanking regions. The remainder of the sequence (963–1566) was similar to ALV (KT030344_AHSV-3_OBP-116_1998) with percentage sequence identity of 98.5%, in (Figure 2B,C). Maximum likelihood analysis of the concatenated region 963–495 had KP939529_AHSV-3_DG25324_14 clustering in the same sub-group as the other serotype 3 vaccine/reference isolates (Figure 2D), while analysis of the region 496–962 clustered KP939529_AHSV-3_DG25324_14 with the sub-group of serotype 5 and the vaccine 8 reassortment (Figure 2E).

### 3.3. Segment-7 (VP7)

Intragenic recombination analysis of more than 100 AHSV segment-7 sequences predicted a single cross-over event at position 642 of sequence KP939765_AHSV-5_86_94. Phylogenetic trees were constructed using Maximum-likelihood predicted KP939765_AHSV-5_86_94 to cluster between the two major groups (Figure 3A). The recombination event had statistical support with *p*-Values of GENECONV *p* = 0.011455, Maxchi *p* = 4.25 × 10^−6^, and Chimaera predicted *p* = 1.71 × 10^−6^. BootScan analysis was performed with vaccine-derived KT71507_AHSV-2_OBP-252_1998 representing the major parent group and wild-type virus sequence KP939764_AHSV-5_47_86 from the minor parental group (Figure 3B). Comparison of sequence similarity between KP939765_AHSV-5_86_94 with the two representatives are indicated in Figure 3C. A high degree of sequence similarity among all three isolates was observed before 642, while after the recombination event isolate KP939765_AHSV-5_86_94 had a significantly higher sequence similarity to KP939764_AHSV-5_47_86 than the vaccine-associated KT71507_AHSV-2_OBP-252_1998. Using maximum likelihood analysis of region 1–625 and region 626–1167 respectively, the N-terminal region of sample KP939765_AHSV-5_86_94 cluster with reference isolates of serotype 2 and 5 (Figure 1D), while the C-terminal region cluster in the other major group consisting of the remaining segment-6 sequences (Figure 1E).

### 3.4. Segment-10 (NS3)

RDP analysis of AHSV predicted a single cross-over event at position 192 of three sequences, reference KP939791_AHSV-5_30_62, KP939793_AHSV-5_42_01, and KP93790_AHSV-5_2_96. These three isolates form a single sub-group A1 in a maximum likelihood phylogenetic analysis (Figure 4A). The parental sequences predicted by RDP belongs to sub-groups A2 and A4 (Figure 4A). The recombination event had statistical support of *p*-values using Siscan of *p* = 1.3 × 10^−15^ and 3Seq of *p* = 4.1 × 10^−3^. BootScan analysis using four reference isolates KP939791_AHSV-5_30_62, KP940106_AHSV-8_10_62, KP939688_AHSV-4_32_62, and KP39920_AHSV-6_2_75 predicted a recombination signal around positions 150–250, but it is not clear which of the sequences were the true recombinant or parental isolates (Figure 4B). Each of these reference isolates shares significant sequence similarity with the other isolates in their sub-groups (A1, A2 and A4), resulting in a similar Bootscan result when other representatives from each sub-group were analyzed (results not shown). The Maximum Likelihood analysis of segment-10 sequences belonging to group alpha, indicated that sub-group A1, A2, and A3 forms part of a separate lineage to sub-group A4 when only region 193–757 are considered (Figure 4D) [38]. In contrast, when region 1–192 was analyzed sub-group A3, a1 and B4 formed a separate lineage to sub-group A2 (Figure 4E).

## 4. Discussion

Recombination has been described in all the major categories of RNA viruses at various frequencies and efficiencies [23]. The rate of recombination is under the influence of the selection pressure exerted on the virus population. It can contribute to virus evolution if a viable phenotype is selected for by changing environmental conditions, or it can produce highly fit viral genomes to rescue low fit parental viruses by repairing fatal mutations in essential genes [20,39]. Similar to mutation and reassortment, recombination can create a condition whereby the most environmentally adapted combinations emerge, leading to an increase in viral survival against host defences [40]. In order for recombination to occur in nature, it is essential that a single host or vector cell is co-infected by two or more viruses. This implies multiple feedings on different viraemic hosts by a single vector, or that the host is infected by multiple vectors with different strains of the same virus. The latter could result from vaccination during active outbreaks

Alignments of individual genomic segments were generated using over 100 AHSV genomes. These sequences included field isolates obtained between 1933 and 2014, OIE-reference strains, and attenuated live viruses (ALV) used in vaccines by Onderstepoort Biological Products (OBP). The alignments were used to identify possible recombination events and four genome segments provided evidence of intragenic recombination. The first genomic recombination event was predicted in Segment-1. The latter consists of 3965 base pairs and is therefore the largest of the AHSV genome segments. It codes for virus protein 1 (VP1), one of the enzymatic minor core proteins responsible for the RNA-dependent-RNA-Polymerase activity [41]. In BTV, a single cross-over event was predicted at around position 1623 of segment-1 between BTV-17/USA and BTV-2/USA isolates resulting in BTV-8/NT [17]. RDP predicted that a single sequence KP939482_AHSV-3_2_89, containing a double cross-over mosaic gene. Sample AHSV-3_2_89 was isolated in 1989 from blood of a vaccinated horse in KwaZulu-Natal, South Africa. Based on phylogenetic analysis, KP939482_AHSV-3_2_89 groups within the major group, yet predictions were that the region between 1524 and 1623bp have recombined with KP939479_AHSV-3_13_63. The latter is the OIE-reference isolate and was used to generate the ALV, which forms the basis of the current commercial vaccine [42]. It was documented that the horse that AHSV-3_2_89 was isolated from was vaccinated, but no additional information exists pertaining to clinical signs in the horse. It is possible that KP939482_AHSV-3_2_89 is a result of recombination between the original ALV of serotype 3 and a field virus. Recent complete genome sequencing of the commercial vaccine virus strains indicated that ALV-AHSV-3 has a reassortment of segment-1 with ALV-AHSV-1 [27]. Previous work using reference isolate AHSV-3_13_63 indicated it was a mixture of serotype 1 and serotype 3 [43]. Based on this information, recombination detected in segment-1 of KP939482_AHSV-3_2_89 could be a result of ALV-3 recombining with field viruses explaining why this event was only detected in a single isolate.

The second genome segment to present evidence of intragenic recombination is Segment-6 encoding VP5, which forms the inner layer of the outer capsid. VP5 has been shown to play a role in membrane destabilization prior to virus entry into cells [44]. The genome segment is 1566bp in size and no recombination was detected in BTV [17]. Clustering of segment-6 within serotypes was expected, since VP5 interacts with VP2 in the outer capsid suggesting co-evolution, with VP2 being the major serotype-specific determinant [14]. The recombination event was predicted in the major group, consisting of serotypes 5, 3, 6, and 9. Recombination with strong statistical support was predicted in isolate KP939529_AHSV-3_DG25324_14 in relation to ALV strains KT030344_AHSV-3_OBP-116_1998 and KT715635_AHSV-8_OBP-252_1998. The sequence obtained from the serotype 8 vaccine (ALV-8) was identified as a reassortment of segment-6 between serotypes 8 and 5 [28]. Since isolate AHSV-3_DG25324_14 possesses a segment-2 sequence identical to ALV-3 (data not shown), it could be assumed that the single isolate recombination event was due to two vaccine viruses recombining.

Segment-7 is conserved, 1167bp in size and codes for the 349 amino acid major core protein VP7 [45]. Maximum likelihood analysis of all the AHSV segment-7 sequences divided the isolates into a small group consisting of reference strains KP939761_AHSV-5_30_62, KP939459_AHSV_2_82_61 and ALVs from serotype 2 and 8, as well as field isolates derived from vaccine viruses. The remainder of the sequences, except for isolate KP939765_AHSV-5_86_94, were grouped in the major cluster. KP939765_AHSV-5_86_94 clustered independently of the two aforementioned groups and was the only isolate to provide evidence of recombination. A single cross-over event around position 642 was predicted with any of the sequences in the small group as parent of the region before the breakpoint and field isolates from the major group as parent for the remainder of the sequence. Since this was a single event, and dual infection with wild type viruses has never been observed, it is again plausible that recombination between a vaccine ALV and a field virus resulted in recombination in segment-7 of KP939765_AHSV-5_86_94. The same genomic segment had a double cross-over predicted at positions 460 and 623 in BTV [17].

The last recombination event appeared in of segment-10 of multiple isolates. Segment-10 is the smallest and second most variable genomic segment, coding for two in frame non-structural proteins involved in virus release NS3/NS3A [46]. In BTV, a double cross-over event was predicted for Segment-10 between bp 158 and 259 of BTV-9 [17]. NS3 is a cytotoxic protein synthesized in small quantities within infected cells and participates in alteration of the cell membrane, virus exit and cell death [47,48]. Phylogenetic analysis of segment-10 produces three major groups, with smaller clusters within each group [38]. A single cross-over event around position 192 was predicted in multiple isolates, all of which were associated with the same group A1 in the alpha cluster. Similarly, the parental sequences originated from multiple isolates belonging to groups A2 and A4 respectively. This points to an ancestral recombination event, which has since manifested in separate lineages. The predicted cross-over sight was in the conserved region of residues 54 to 58 [38].

This study provides the first evidence of intragenic recombination in African horse sickness virus, albeit less abundant than in bluetongue virus. In comparison, BTV has more antigenic diversity than AHSV with an ever-growing number of serotypes described, 27 and nine respectively [49]. The distribution of BTV includes every continent except Antarctica, while AHSV is endemic to sub-Saharan Africa with sporadic escapes to North Africa, the Middle East, and Mediterranean countries [50]. Genomes showing intragenic recombination events in BTV became the predominant field strains, contribution to genetic diversity and influencing its epidemiology [17]. In contrast, the majority of intragenic recombination observed in AHSV resulted because of exchanges between live attenuated vaccine viruses and field isolates. Annual vaccination of horses using the trivalent and tetravalent vaccine from OBP is compulsory in South Africa [51]. It is possible that the high incidence of recombination detected between ALVs and field viruses is an artefact of the recombination detection method or virus isolation method employed. The OBP vaccine contains ALV strains derived from viruses isolated in the 1960s and therefor significant sequence diversity was observed between the reference and ALV strains on the one hand and recent field isolates on the other. Since the majority of the viruses were isolated in South Africa or neighbouring countries, the small geographic distribution of isolates might contribute to the high percentage nucleotide identity observed. Phylogenetic analysis of individual genome segments of AHSV demonstrated that the majority of sequences group in temporal distribution rather than by serotype, with the exceptions of segment-2 (VP2) and segment-5 (VP5). This implies that co-circulating viruses are similar to one another, inhibiting the detection of recombination events that might have occurred between field isolates. Therefore, the only recombination detected by the methods used in this study was between genetically diverse older and recent samples, where the older samples were used in ALV development. The sequences used in this study were from complete genomes of viruses isolated on cell culture, it is plausible that the recombination events involving ALV viruses provided the hybrid viruses with a selection advantage to cell culturing, rather than an increase in virulence or viral fitness.

Unfortunately, the intragenic recombination described here cannot account for the significant incongruence observed between phylogenetic trees generated from different AHSV segments. This necessitates subsequent investigations into the role of reassortment on AHSV evolution.

## Figures and Tables

**Figure 1 viruses-11-00654-f001:**
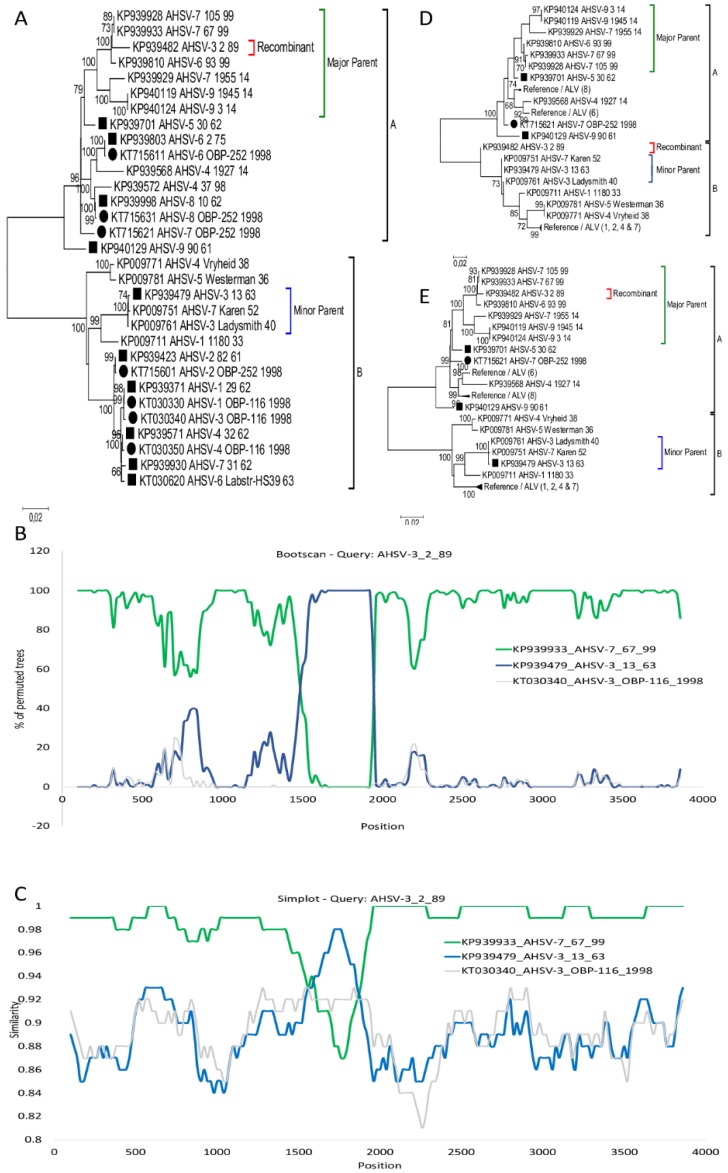
Evidence of intragenic recombination in Segment-1 of KP939482_AHSV-3_2_89. (**A**) ML phylogenetic tree showing the relationship of segment-1 in selected African horse sickness virus (AHSV) isolates. Reference isolates are indicated with a square and attenuated live virus (ALV) sequences with a circle. (**B**) BootScan results of KP939482_AHSV-3_2_89 with its parents KP939933_AHSV-7_67_99 and KP939479_AHSV-3_13_62 and an outgroup KT030340_AHSV-3_OBP-116_1998. (**C**) Comparison of the sequence similarity between KP939482_AHSV-3_2_89 and isolates KP939933_AHSV-7_67_99, KP939479_AHSV-3_13_62 and KT030340_AHSV-3_OBP-116_1998 using SimPlot. (**D**) Phylogenetic tree of the concatenated segment-1 sequence of region 1865–1523 of selected AHSV isolates. (**E**) Phylogenetic tree of region 1524–1864 from segment-1 of selected AHSV isolates.

**Figure 2 viruses-11-00654-f002:**
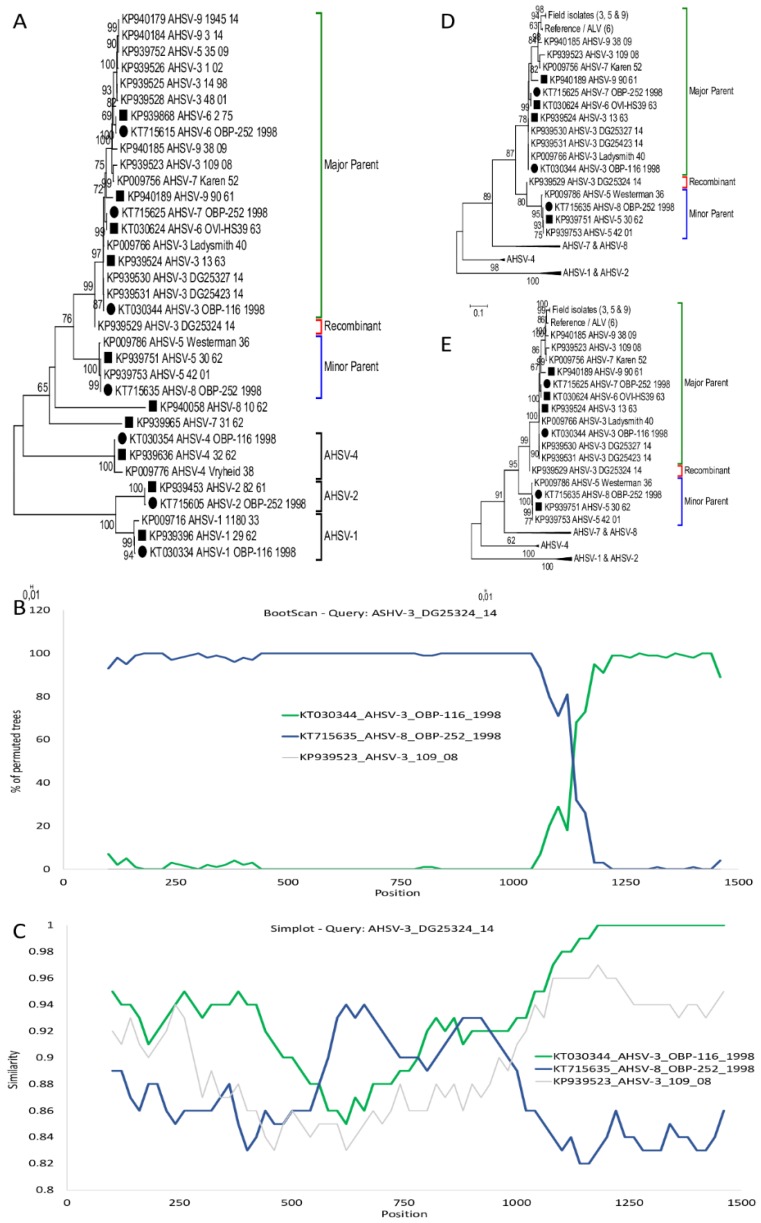
Evidence of intragenic recombination in Segment-6 of KP939529_AHSV-3_DG25321_14. (**A**) ML phylogenetic tree showing the relationship of segment-6 using selected African horse sickness virus (AHSV) isolates. Reference isolates are indicated with a square, while attenuated live virus (ALV) sequences are marked with a circle. (**B**) BootScan results of KP939529_AHSV-3_DG25321_14 with parental isolates KT030344_AHSV-3_OBP-116_1998 and KT715635_AHSV-8_OBP-252_1998, as well as an outgroup KP939523_AHSV-3_109_08. (**C**) Comparison of the sequence similarity between KP939529_AHSV-3_DG25321_14 and isolates KT030344_AHSV-3_OBP-116_1998, KT715635_AHSV-8_OBP-252_1998 and KP939523_AHSV-3_109_08 using SimPlot. (**D**) Phylogenetic tree of the concatenated segment-6 sequence of region 963–495 of selected AHSV isolates. (**E**) Phylogenetic tree of region 496–962 from segment-6 of selected AHSV isolates.

**Figure 3 viruses-11-00654-f003:**
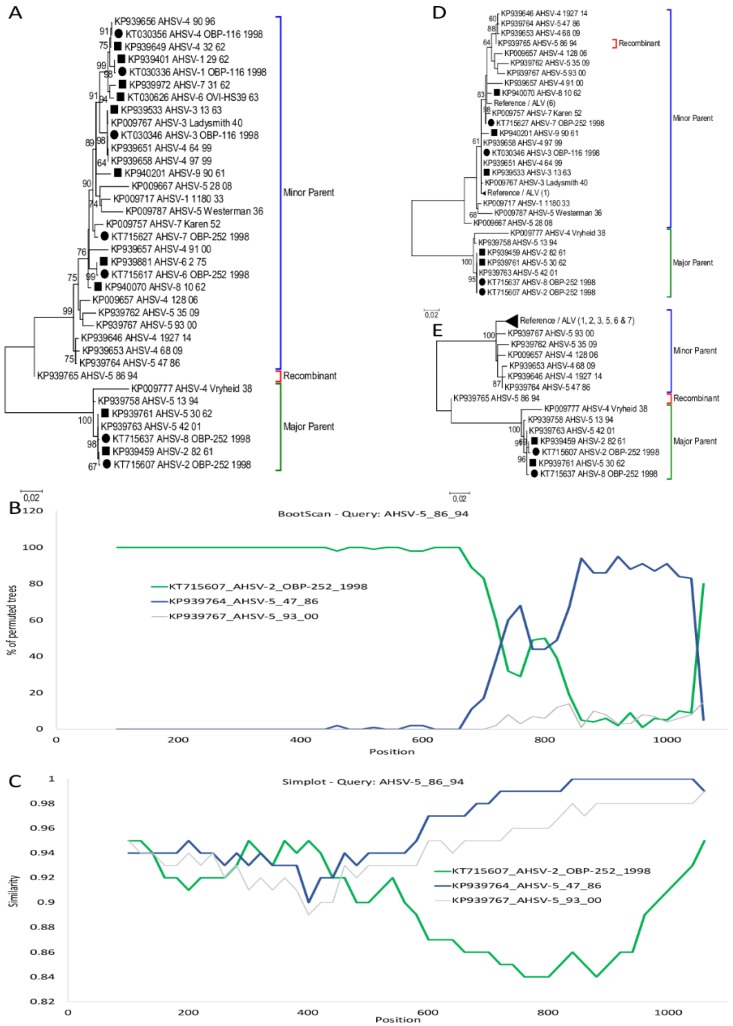
Evidence of intragenic recombination in Segment-7 of KP939765_AHSV-5_86_94. (**A**) ML phylogenetic tree showing the relationship of segment-7 using selected African horse sickness virus (AHSV) sequences. Reference isolates are indicated with a square, while attenuated live virus (ALV) sequences are displayed with a circle. (**B**) BootScan results of KP939765_AHSV-5_86_94 with possible parents KT715607_AHSV-2_OBP-252_1998 and KP939764_AHSV-5_47_86 and an outgroup KP939767_AHSV-5_93_00. (**C**) Comparison of the sequence similarity between KP939765_AHSV-5_86_94 and isolates KT715607_AHSV-2_OBP-252_1998, KP939764_AHSV-5_47_86 and KP939767_AHSV-5_93_00 using SimPlot. (**D**) Phylogenetic tree of segment-7 sequence of region 1–642 of selected AHSV isolates. (**E**) Phylogenetic tree of region 642–1105 of segment-7 from selected AHSV isolates.

**Figure 4 viruses-11-00654-f004:**
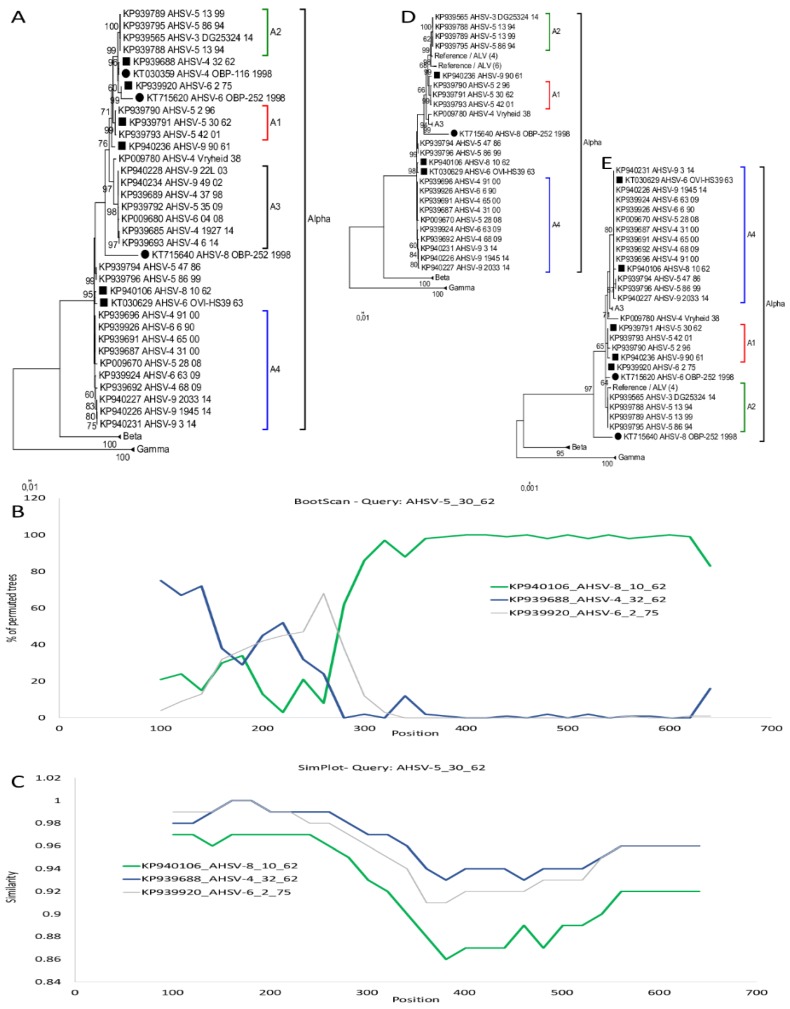
Predicted intragenic recombination in Segment-10 of KP939791_AHSV-5_30_62 (B2). (**A**) ML phylogenetic tree showing the relationship of segment-10 using selected African horse sickness virus (AHSV) sequences. Reference isolates are indicated with a square and attenuated live viruses (ALVs) with a circle. (**B**) BootScan results of KP939791_AHSV-5_30_62 with possible parents KP940106_AHSV-8_10_62 and KP39688_AHSV-4_32_62 and an outgroup KP939920_AHSV-6_2_72. (**C**) Comparison of the sequence similarity between KP939791_AHSV-5_30_62 and isolates KP940106_AHSV-8_10_62, KP39688_AHSV-4_32_62 and KP939920_AHSV-6_2_72 using SimPlot. (**D**) Phylogenetic tree of segment-10 sequence of region 193–757 of selected AHSV isolates. (**E**) Phylogenetic tree of region 1–192 of segment-10 from selected AHSV isolates.

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
