# Peer review of "Evidence of Intragenic Recombination in African Horse Sickness Virus"

_viruses, 2019, doi:10.3390/v11070654_

Round 1

Reviewer 1 Report

Whole genome sequencing and analyses of AHSVs is important, interesting and still in its infancy compared to other viruses. It is also of interest to document evidence of the frequency of reassortment and/or intragenic recombination between vaccine and field strains.

For a better overview please consider adding a Table summarizing the basic information of at least the 87 newly reported sequences here by their serotype, country of origin, date of isolation, Genbank number, and a column indicating genome segments showing intragenic recombination.

The Discussion on intragenic recombination in genome segment 10(NS3) needs to be improved and re-interpreted. Of importance is to interpret and assess the findings by taking into account the paper by Quant and co-workers (2008)  on 145 AHSV GS10(NS3) sequences which is not mentioned at all.

Bring the "four phylogentic groups" mentioned in line 313-314 in context with the known NS3 clades, alpha, beta and gamma. Also update Fig 4. to indicate the NS3 clades.

Please be more specific as to the position of the cross over in context of the known domains in NS3, between the TM sites or where?

Quan M, Van Vuuren M, Howell PG, Groenewald D, Guthrie AJ (2008) Molecular epidemiology of the African horse sickness virus S10 gene. J Gen Virol 89:1159–1168

"Between 2004 and 2006, 145 African horse sickness viruses (AHSV) were isolated from blood
and organ samples submitted from South Africa to the Faculty of Veterinary Science, University of
Pretoria. All nine serotypes were represented, with a range of 3–60 isolates per serotype. The
RNA small segment 10 (S10) nucleotide sequences of these isolates were determined and the
phylogeny investigated. AHSV, bluetongue virus (BTV) and equine encephalosis virus (EEV) all
formed monophyletic groups and BTV was genetically closer to AHSV than EEV. This study
confirmed the presence of three distinct S10 phylogenetic clades (alpha, beta and gamma). Some serotypes (6, 8 and 9 in alpha; 3 and 7 in beta; 2 in gamma) were restricted to a single clade, while other serotypes (1, 4 and 5) clustered into both the alpha and gamma clades..... "

There are a few grammatical errors to be corrected in lines:

38    double-stranded

40    .. molecular mass

53    occurs

89    fits

103-138    adjust the right margin

181    form a

186    it is not

255    segments

331    1960s

346    can not

Author Response

Response to comments, reviewer 1

The Discussion on intragenic recombination in genome segment 10(NS3) needs to be improved and re-interpreted. Of importance is to interpret and assess the findings by taking into account the paper by Quant and co-workers (2008)  on 145 AHSV GS10(NS3) sequences which is not mentioned at all.

Bring the "four phylogentic groups" mentioned in line 313-314 in context with the known NS3 clades, alpha, beta and gamma. Also update Fig 4. to indicate the NS3 clades.

- The Qaun et al., 2008 reference [37] was added and this work was referred to in Section 3.4 and in the Discussion. This was also included in Figure 4.

Please be more specific as to the position of the cross over in context of the known domains in NS3, between the TM sites or where?

- Text was included in the Discussion, referring to Quan et al., 2008, indicating that the predicted cross-over sight was in the conserved region of the gene, between residues 54 to 58.

Quan M, Van Vuuren M, Howell PG, Groenewald D, Guthrie AJ (2008) Molecular epidemiology of the African horse sickness virus S10 gene. J Gen Virol 89:1159–1168

"Between 2004 and 2006, 145 African horse sickness viruses (AHSV) were isolated from blood and organ samples submitted from South Africa to the Faculty of Veterinary Science, University of Pretoria. All nine serotypes were represented, with a range of 3–60 isolates per serotype. The RNA small segment 10 (S10) nucleotide sequences of these isolates were determined and the phylogeny investigated. AHSV, bluetongue virus (BTV) and equine encephalosis virus (EEV) all formed monophyletic groups and BTV was genetically closer to AHSV than EEV. This study confirmed the presence of three distinct S10 phylogenetic clades (alpha, beta and gamma). Some serotypes (6, 8 and 9 in alpha; 3 and 7 in beta; 2 in gamma) were restricted to a single clade, while other serotypes (1, 4 and 5) clustered into both the alpha and gamma clades..... "

-       Figure 4 and the text were changed to include to nomenclature as described in Quan et al., 2008.

There are a few grammatical errors to be corrected in lines:

38    double-stranded

40    .. molecular mass

53    occurs

89    fits

103-138    adjust the right margin

181    form a

186    it is not

255    segments

331    1960s

346 can not

- All corrections were made as indicated.

Reviewer 2 Report

Review of viruses-484551

In the manuscript “Evidence of intragenic recombination in African horse sickness virus” the authors set out to document evidence for intrasegmental recombination in African horse sickness virus (AHSV), and estimate the influence of these event on the evolution of the virus. They are ideally situated at the Onderstepoort Veterinary Institute in South Africa to do this, as they have access there to all the historical and reference AHSV strains plus a large set of field isolates.  For this study they determined the complete genome sequences of 87 isolates, and supplemented this with additional data from GenBank. They then searched for evidence of intragenic recombination, which they identified in four of the genome segments as single events. Based on the data presented here intragenic recombination does not appear to play a major role as driver of genetic diversity in AHSV, or in the evolution of epidemiology of the virus.

Specific comments (in chronological order)

1.  Correct spelling of “reassortment” in line 15.

2.  Standardise spelling of the full name of AHSV to the correct (ICTV approved) version of the name as used in the title and line 14, but not as in lines 22 and 25.

3.  This is a personal opinion, and used differently by different authors, but I do not agree with saying that the AHSV genome encodes five non-structural proteins (line 41). There are four different non-structural proteins. NS3A is just a slightly truncated (11 amino acids shorter) version of NS3, but they essentially represent the same protein, translated from two different in-frame start codons on the same genome segment.

4.  In the introduction (and throughput most of the paper) the authors use the concept of recombination in the “narrow” sense, specifically referring only to homologous recombination from crossover events (line 48). However in the abstract (line 15) they use it in the “wide” sense, including both reassortment of genome segments and homologous intragenic recombination. In dsRNA viruses with segmented genomes reassortment is a common event, while intragenic recombination, as also illustrated in the paper, seems to be rare and of minor importance for the virus’ evolution. As the standard use of the term “recombinant AHSV” will often refer to a progeny virus representing a genomic reassortant with genome segments from two different parents, I prefer the use of the “wide” sense application of the concept recombination. They will need to choose which application to support, and do so throughout.

5.  In both line 56 & line 60 of the introduction the authors state that “there are no reports on the influence of genetic recombination in AHSV”. Closer scrutiny of the publication by Weyer et al (2016) (ref #24 in their reference list) shows that those authors did search for intrasegment recombination in full genome sequences of 55 AHSV isolates using the same RDP software utilised in this study, but detected no evidence of intragenic recombination.

6.  In my opinion the statement “Chimeric genes generated through intragenic recombination in BTV have resulted in these viruses becoming the prevailing strain” in lines 57-58 of the introduction represents on over-interpretation of the data from He et al (2010), and creates the wrong impression with respect to the potential importance of these events in orbiviruses. The same for lines 324-325 in the discussion.

7.  Linked to my previous comment, the only reference made by the authors to this kind of work done in related viruses is to the He et al (2016) paper, the sole publication where this phenomenon was investigated in bluetongue virus (BTV). There are however a number of papers on this topic for rotavirus, also a member of the family Reoviridae. These need to be included in the introduction and/or discussion. Of note is R.J. Woods / Infection, Genetics and Evolution 32 (2015) 354–360, who analysed the 11 genome segments of 797 rotavirus strains, and his conclusion that there is no evidence that intrasegmental recombination leads to ongoing transmission or plays a constructive role in rotavirus evolution.

8.  Details of the “more than 100 AHSV isolates” (line 65) used in this study need to be provided in a Table either in Materials & Methods or as Supplementary Material. The GenBank reference numbers for the gene sequences are not sufficient and difficult to use. The table could provide information on which isolates represent reference strains, ALV vaccine strains, field isolates, dates of isolation, passage history, etc. As all figures only include a selected subset of sequences, with some of these details included there, a full list is required. Where possible information should also be provided on what method was used to determine or confirm the serotype of the isolate. Lane 77 states that cells “were infected with freeze-dried individual AHSV isolates”. However in the discussion it is mentioned that some isolates contained a mixture of serotypes (e.g. line 275), and further they speculate that some of the recombination detected can be “an artefact of the …. virus isolation method employed.” (line 330). They only detected four recombination events in all their analyses, and should some of those then be artefacts, this further illustrates the limited impact of intragenic recombination in AHSV.

9. The abbreviation for attenuated live virus (ALV) should be provided in line 71, not line 257.

10. The nomenclature for describing the genome segments needs to be standardised throughout. They use all possible variants at different places in the manuscript – e.g. Segment 1, segment 1, Segment-1, segment-1, Seg-1 or seg-1. Different authors have different preferences, however at least for BTV the current opinion favours Seg-1.

11.  In line 296, KP939765_AHSV-5_86_84 should be corrected to KP939765_AHSV-5_86_94.

12.  For the recombination event in Seg-7, the data indicate that any sequence from the small group could have acted as parent for the region before the breakpoint. The small group contains two reference strains, two ALV strains, and three field isolates. What additional evidence supports the conclusion that “recombination between a vaccine ALV and field isolates resulted in recombination in segment 7” (lines 302-303)? This also impacts on the statement in lines 326-327 of the discussion.

13.  The argument around BTV’s antigenic diversity (lines 318-323) as evidence for its greater need for genome-wide genetic diversification compared to AHSV is purely speculative.  There are many papers quantifying and comparing the degree of intra-serogroup diversity per gene for these two viruses, so actual data can be used here.

14.  Please clarify how the recombination detection method (line 330) could result in the high incidence of recombination being an artefact. If so, what could be done to verify or rectify the problem?  

15.  The concluding paragraph is confusing. What do the authors mean by “the significant incongruence observed between AHSV segments”? Some genome segments show close to 100% nucleotide identity across all strains, whereas other segments are highly variable, resulting from the combined effects of mutation and selection. Furthermore, it is well known that reassortment can occur during a mixed AHSV infection, hence only Seg-2 (and to a lesser extent Seg-6) phylogenies show clustering based on serotype.

Author Response

Response to comments, reviewer 2

1.  Correct spelling of “reassortment” in line 15.

- Correction made

2.  Standardise spelling of the full name of AHSV to the correct (ICTV approved) version of the name as used in the title and line 14, but not as in lines 22 and 25.

- African horse sickness virus is used consistently

3.  This is a personal opinion, and used differently by different authors, but I do not agree with saying that the AHSV genome encodes five non-structural proteins (line 41). There are four different non-structural proteins. NS3A is just a slightly truncated (11 amino acids shorter) version of NS3, but they essentially represent the same protein, translated from two different in-frame start codons on the same genome segment.

- Changes were made to only refer to four non-structural proteins.

4.  In the introduction (and throughput most of the paper) the authors use the concept of recombination in the “narrow” sense, specifically referring only to homologous recombination from crossover events (line 48). However in the abstract (line 15) they use it in the “wide” sense, including both reassortment of genome segments and homologous intragenic recombination. In dsRNA viruses with segmented genomes reassortment is a common event, while intragenic recombination, as also illustrated in the paper, seems to be rare and of minor importance for the virus’ evolution. As the standard use of the term “recombinant AHSV” will often refer to a progeny virus representing a genomic reassortant with genome segments from two different parents, I prefer the use of the “wide” sense application of the concept recombination. They will need to choose which application to support, and do so throughout.

- A distinction between recombination (intragenic and reassortment) was included in the text.

5.  In both line 56 & line 60 of the introduction the authors state that “there are no reports on the influence of genetic recombination in AHSV”. Closer scrutiny of the publication by Weyer et al (2016) (ref #24 in their reference list) shows that those authors did search for intrasegment recombination in full genome sequences of 55 AHSV isolates using the same RDP software utilised in this study, but detected no evidence of intragenic recombination.

- Since ref#24 didn’t find any evidence of intragenic recombination, the statement that no reports on the influence of homologous recombination is true.

6.  In my opinion the statement “Chimeric genes generated through intragenic recombination in BTV have resulted in these viruses becoming the prevailing strain” in lines 57-58 of the introduction represents on over-interpretation of the data from He et al (2010), and creates the wrong impression with respect to the potential importance of these events in orbiviruses. The same for lines 324-325 in the discussion.

- The text at the end of the Introduction was changed to remove the erroneous interpretation and only report on the actual findings.

7.  Linked to my previous comment, the only reference made by the authors to this kind of work done in related viruses is to the He et al (2016) paper, the sole publication where this phenomenon was investigated in bluetongue virus (BTV). There are however a number of papers on this topic for rotavirus, also a member of the family Reoviridae. These need to be included in the introduction and/or discussion. Of note is R.J. Woods / Infection, Genetics and Evolution 32 (2015) 354–360, who analysed the 11 genome segments of 797 rotavirus strains, and his conclusion that there is no evidence that intrasegmental recombination leads to ongoing transmission or plays a constructive role in rotavirus evolution.

- The report of Woods (2015)on rotaviruses was cited as well as the work of Quan et al., (2008) on AHSV.

8.  Details of the “more than 100 AHSV isolates” (line 65) used in this study need to be provided in a Table either in Materials & Methods or as Supplementary Material. The GenBank reference numbers for the gene sequences are not sufficient and difficult to use. The table could provide information on which isolates represent reference strains, ALV vaccine strains, field isolates, dates of isolation, passage history, etc. As all figures only include a selected subset of sequences, with some of these details included there, a full list is required. Where possible information should also be provided on what method was used to determine or confirm the serotype of the isolate. Lane 77 states that cells “were infected with freeze-dried individual AHSV isolates”. However in the discussion it is mentioned that some isolates contained a mixture of serotypes (e.g. line 275), and further they speculate that some of the recombination detected can be “an artefact of the …. virus isolation method employed.” (line 330). They only detected four recombination events in all their analyses, and should some of those then be artefacts, this further illustrates the limited impact of intragenic recombination in AHSV.

- An additional table was included showing the information on origin and passage history. This could be included as supplementary information.

9. The abbreviation for attenuated live virus (ALV) should be provided in line 71, not line 257.

- The explanation of the abbreviation was moved up to its first occurrence.

10. The nomenclature for describing the genome segments needs to be standardised throughout. They use all possible variants at different places in the manuscript – e.g. Segment 1, segment 1, Segment-1, segment-1, Seg-1 or seg-1. Different authors have different preferences, however at least for BTV the current opinion favours Seg-1.

- The text was changed to use “segment-1” throughout.

11.  In line 296, KP939765_AHSV-5_86_84 should be corrected to KP939765_AHSV-5_86_94.

- The correction was done.

12.  For the recombination event in Seg-7, the data indicate that any sequence from the small group could have acted as parent for the region before the breakpoint. The small group contains two reference strains, two ALV strains, and three field isolates. What additional evidence supports the conclusion that “recombination between a vaccine ALV and field isolates resulted in recombination in segment 7” (lines 302-303)? This also impacts on the statement in lines 326-327 of the discussion.

- An explanation was added to the text that dual infections are not expected to be with two wild type strain, and this is only likely when ALVs are introduced during natural infection.

13.  The argument around BTV’s antigenic diversity (lines 318-323) as evidence for its greater need for genome-wide genetic diversification compared to AHSV is purely speculative.  There are many papers quantifying and comparing the degree of intra-serogroup diversity per gene for these two viruses, so actual data can be used here.

- The sentence referred to here, “Accounting for antigenic diversity, wide distribution as well as the broader host range BTV has in comparison to AHSV, it is evident that BTV is under higher selection pressure and thus has a greater need for genetic diversification.” was deleted to remove the speculation.

14.  Please clarify how the recombination detection method (line 330) could result in the high incidence of recombination being an artefact. If so, what could be done to verify or rectify the problem?  

- Intragenic recombination prediction is inhibited by low sequence diversity. Since the highest sequence diversity is due to genetic drift and thus between old and new isolates, it follows that there will be higher diversity between vaccine (or vaccine-derived) viruses and new field isolates.

15.  The concluding paragraph is confusing. What do the authors mean by “the significant incongruence observed between AHSV segments”? Some genome segments show close to 100% nucleotide identity across all strains, whereas other segments are highly variable, resulting from the combined effects of mutation and selection. Furthermore, it is well known that reassortment can occur during a mixed AHSV infection, hence only Seg-2 (and to a lesser extent Seg-6) phylogenies show clustering based on serotype.

- Additions were made to the text of the final paragraph to indicate that the incongruence refers to the different trees constructed using data from different genome segments.